# Efficacy of Multimodal Analgesia for Postoperative Pain Management in Head and Neck Cancer Patients

**DOI:** 10.3390/cancers13061266

**Published:** 2021-03-12

**Authors:** Ashley Hinther, Steven C. Nakoneshny, Shamir P. Chandarana, T. Wayne Matthews, Robert Hart, Christiaan Schrag, Jennifer Matthews, C. David McKenzie, Gordon H. Fick, Joseph C. Dort

**Affiliations:** 1Section of Otolaryngology—Head and Neck Surgery, Department of Surgery, Cumming School of Medicine, University of Calgary, Calgary, AB T2N 4Z6, Canada; avhinthe@ucalgary.ca (A.H.); shamir.chandarana@ucalgary.ca (S.P.C.); wmatthew@ucalgary.ca (T.W.M.); Robert.Hart@albertahealthservices.ca (R.H.); 2Ohlson Research Initiative, Arnie Charbonneau Cancer Institute, Cumming School of Medicine, University of Calgary, Calgary, AB T2N 4Z6, Canada; scnakone@ucalgary.ca; 3Section of Plastic and Reconstructive Surgery, Department of Surgery, Cumming School of Medicine, University of Calgary, Calgary, AB T2N 4Z6, Canada; cschrag@me.com (C.S.); jennifer.matthews1@me.com (J.M.); cdavidmckenzie@mac.com (C.D.M.); 4Community Health Sciences, Cumming School of Medicine, University of Calgary, Calgary, AB T2N 4Z6, Canada; ghfick@ucalgary.ca

**Keywords:** multimodal analgesia, perioperative pain control, postoperative pain, head and neck surgery, head and neck cancer

## Abstract

**Simple Summary:**

Chronic opioid use is a serious global health problem and surgery is often the point of initial exposure for many chronic opioid users. Multimodal analgesia (MMA) is an approach designed to reduce or eliminate opioid use in surgical patients, but it has not been studied in patients having major head and neck surgery. This study explores the impact of an MMA protocol on opioid prescribing practices in patients having major head and neck surgery with flap reconstruction. The results of this study will provide evidence to inform and guide pain management practice in this important patient population. The results of this study could also be applied to other areas of otorhinolaryngology. MMA is an important tool in the effort to reduce chronic opioid use.

**Abstract:**

Postoperative opioid use has been linked to the subsequent development of opioid dependency. Multimodal analgesia (MMA) can reduce the use of opioids in the postoperative period, but MMA has not been well-studied after major head and neck surgery. Our goal is to explore the association between MMA and postoperative opioid use and pain control in patients undergoing major head and neck surgery. We performed a retrospective study in adult (age ≥ 18 years) patients undergoing primary head and neck cancer resection with free-flap reconstruction. All patients were treated using an established care pathway. The baseline group was treated between January 2015–December 2015 (*n* = 41), prior to the implementation of MMA, and were compared to an MMA-treated cohort treated between December 2017–June 2019 (*n* = 97). The primary outcome was the proportion of opioids prescribed and oral morphine equivalents (OMEs) consumed during the hospitalization. The secondary outcome was pain control. We found that the post-MMA group consumed fewer opioids in the postoperative period compared to the pre-MMA group. Prior to post-operative day (POD) 6, pain control was better in the post-MMA group; however, the pain control lines intersect on POD 6 and the pre-MMA group appeared to have better pain control from PODs 7–10. In conclusion, our data suggest MMA is an effective method of pain control and opioid reduction in patients undergoing surgery for head and neck cancer with free flap reconstruction. MMA use was associated with a significant decrease in the quantity of opioids consumed postoperatively. The MMA protocol was associated with improved pain management early in the postoperative course. Finally, the MMA protocol is a feasible method of pain control and may reduce the adverse side effects associated with opioid use.

## 1. Introduction

Adequate postoperative pain control is imperative for successful recovery and rehabilitation following surgery [1]. Pain is prevalent in over 50% of cancer patients with the highest prevalence in patients with head and neck cancer (70%) [2]. Head and neck cancer patients undergoing surgical resection with free-flap reconstruction represent a unique population; patients experience pain secondary to surgery as well as postoperative interventions including nasogastric and tracheotomy tubes [3]. Pain management following major head and neck surgery therefore represents a complex challenge, wherein opioids have traditionally been a mainstay in management [4].

Opioids have a significant side effect profile, including postoperative nausea and vomiting, constipation, sedation, and impaired mobilization. These side effects create further barriers to patients’ postoperative recovery. There is also growing evidence showing an association between opioid use in the acute postoperative period and the subsequent development of chronic opioid use [5,6,7,8,9]. Chronic postoperative opioid use has personal and societal impacts and is a major contributor to the current opioid crisis [10]. We recently investigated our centre’s effectiveness in managing postoperative pain in head and neck cancer patients undergoing free flap reconstruction. Historically, our centre treated pain after major head and neck cancer surgery with opioids. However, despite the preponderant use of opioids, our patients’ pain was not optimally managed [4]. Given these results, we believe multimodal analgesia (MMA) may be an effective method of postoperative pain management in head and neck cancer surgery.

An enhanced recovery after surgery (ERAS) guideline for head and neck cancer patients addresses 17 topic areas within the preoperative, intraoperative, and postoperative care time periods [11]. Effective pain management is an important goal of the ERAS protocol and uses multimodal analgesia (MMA) to accomplish this goal. MMA is defined as the concurrent use of more than one modality of pain control to achieve effective analgesia. MMA reduces opioid consumption and may reduce opioid-related side effects [11,12]. A recent study by Vu et al. showed MMA was effective in reducing opioid consumption in the post-anesthesia care unit (PACU) in patients undergoing major head and neck surgery [13]. Many current MMA regimens for head and neck cancer surgery patients recommend the combined use of non-steroidal anti-inflammatories (NSAIDs) and/or COX-inhibitors, and paracetamol. Some MMA protocols also include a gabapentinoid to further improve the analgesic effect. Opioids are then reserved for uncontrolled pain and are only considered if MMA approaches are insufficient [11]. Many surgical specialties, including general surgery, gynecology, neurosurgery, and plastic surgery, have implemented MMA protocols in an effort to decrease the proportion of opioids used in the postoperative period and subsequently decrease the negative side effects [14,15,16].

Our centre recently implemented a head and neck ERAS protocol which includes an MMA order set for managing postoperative pain [11]. The Calgary protocol uses a combination of ibuprofen, acetaminophen, and gabapentin and is provided to all patients who have no contraindications to the constituent medications. The primary objective of this study is to assess the proportion and quantity of opioids used prior to and after the implementation of the MMA protocol in patients undergoing major head and neck surgery with free flap reconstruction. Secondarily, we compared the effectiveness of pain control in a predominantly opioid treated group to an MMA treated group. We hypothesized the MMA regimen would reduce the proportion and quantity of opioids consumed and still provide adequate pain control in this challenging patient population. The findings from this research will provide new information that could assist other clinicians in managing postoperative pain in head and neck cancer patients.

## 2. Materials and Methods

### 2.1. Patient Selection

Data were prospectively collected, and retrospectively analyzed, on adult patients (age ≥ 18 years) undergoing primary head and neck cancer resection with free-flap reconstruction at a tertiary, academic head and neck surgical oncology program (University of Calgary, Calgary, Alberta, Canada). Eligible patients were identified from a prospectively annotated database (Otobase^TM^). Two patient groups were compared. A pre-MMA cohort that included patients who underwent surgery between January 2015–December 2015 and were managed using the Calgary Head and Neck Enhanced Recovery Program (CHERP) protocol [4,17,18]. The MMA cohort included all patients who underwent surgery between December 2017–June 2019 and received the CHERP protocol plus additional ERAS elements including treatment with MMA. Figure 1 is a Consolidated Standards of Reporting Trials diagram describing the cohort.

### 2.2. Data Collection

Extracted data included patient demographics, smoking and drinking habits, clinical tumor stage, pathological tumor stage, type of surgery, and adjuvant treatment. The hospital EMR (Sunrise Clinical Manager, Eclipsys Corporation, Atlanta, GA, USA) provided data for pain control (1–10 Numeric Rating Scale (NRS)), as well as the type, timing, and frequency of all analgesics administered. Opioid consumption (primary outcome) was measured in two ways: proportion of pain medications that were opioids and quantity of opioids administered. To simplify comparison of opioid amounts, opioid doses were converted to oral morphine equivalents (OMEs). Pain intensity (secondary outcome) was measured using the NRS. Additional outcomes measured were time to mobilization, length of hospital stay, and complications. Complications were graded using the Clavien-Dindo classification system.

### 2.3. Statistical Analysis

Data management and analyses were carried out using Stata version 16 (Stata Corp, College Station, TX, USA). The analysis was directed by a biostatistician who was a collaborator on the project (GHF). Categorical variables are displayed as proportions and continuous variables are reported as means or medians along with measures of variation as appropriate. Categorical variables are compared using a chi square or Fisher’s exact test and continuous variables are compared using appropriate parametric or non-parametric analyses. Count data (length of stay, time to first mobilization) were analyzed using a truncated negative binomial regression approach. Statistical significance is indicated by a *p*-value of less than 0.05.

Opioid and pain score data were complex and required model-based analytical approaches. Therefore, much of the analysis in this paper is model-based and are forms of logistic regression. The logit (log odds) is considered to be conditional on subjects and one then considers the regression coefficients as being assumed common to subjects. The estimates of the regression coefficients are then said to be adjusted for subjects. These conditional models are also traditionally called mixed models. We used the command melogit (Stata version 16) which provides estimates of the regression coefficients and predictions of the subject components. For example, we presented models for the log of the odds of receiving opioids as functions of the intervention (MMA: yes/no) and indicators for the day of the trial (postoperative day). We further studied a number of models that allowed us to consider a number of potential modifiers/confounders such as age, sex, tumor stage, and others. On the balance of many issues, we decided not to include any additional confounders or modifiers in our reported results. Therefore, only adjustments for within-subject variation were performed. A similar approach was used to analyze the pain score data.

### 2.4. Ethical Approval

The authors used A Project Ethics Community Consensus Initiative (ARECCI) framework to assess for and mitigate ethical risks, including the ARECCI Ethics Screening Tool and the ARECCI Ethics Guidelines. The project was deemed a quality improvement initiative with a minimal risk (ARECCI score = 1) [19].

## 3. Results

Table 1 shows the clinical characteristics of the pre-MMA and post-MMA groups. A total of 41 patients (mean age [SD] 61.2 years [12.3]; 32 [78%] male) were included in the pre-MMA group and 97 (mean age [SD] 61.9 years [11.9]; 66 [68%] male) patients were included in the post-MMA group. The two groups were similar with regard to age, gender, comorbidities, social habits, and tumor characteristics. Oral cavity cancer was the most common indication for surgery in both the pre- and post-MMA groups (56% and 57%, respectively).

Daily opioid consumption as a proportion of total analgesics given is shown in Figure 2 and reveals that fewer opioids were consumed in the post-MMA group. As expected, the proportion of opioids used daily trended downward over the duration of hospitalization in both groups. Mixed-effects logistic regression revealed these inter-group differences were significant.

Patients in the post-MMA group consumed fewer opioids over the duration of their hospital stay. Mean daily opioid consumption was 29.7 mg (SD 5.0) in the post-MMA group versus 43.3 mg (SD 18.8) in the pre-MMA group (*p* = 0.04 Wilcoxon). Figure 3 shows mean daily opioid consumption for the first 14 postoperative days. Interestingly, daily OME consumption is slightly lower in the pre-MMA group for the first 4 postoperative days. After postoperative day 4, OME consumption is much higher in the pre-MMA group. The overall trend for opioid consumption increases over the duration of hospital stay, particularly in the pre-MMA group.

Figure 4 illustrates the proportion of patients with poorly controlled pain on each postoperative day. Poorly controlled pain is defined as a pain score greater than 3 [20,21]. Prior to POD 6 pain control was better in the post-MMA group. The pain control lines cross on POD 6 and the pre-MMA group has better pain control from postoperative days 7–10. After POD 10, the two groups have comparable pain scores. As expected, the overall trend in pain scores for both groups decreases over the duration of hospital stay. Mixed-effects logistic regression revealed that the demonstrated differences were significant.

We also evaluated time to first mobilization, length of stay, and overall complications (measured by Clavien-Dindo classification). These data are reported in Table 2. Patients in the post-MMA group were mobilized earlier than those in the pre-MMA group (*p* < 0.001 Wilcoxon). Neither complications nor length of stay were different between the two groups.

## 4. Discussion

This study shows that MMA is associated with a lower proportion of opioid use and lower opioid consumption as measured by OMEs in patients undergoing surgery for head and neck cancer with free flap reconstruction. Furthermore, MMA use was also associated with lower pain scores early in the postoperative period (POD 0–6). The pre- and post-MMA cohorts had comparable rates of postoperative complications suggesting MMA use is not associated with increased complications. MMA, therefore, appears to be a feasible method of pain control for postoperative head and neck cancer patients and may reduce the adverse side effects associated with opioid use.

Postoperative opioid use has been linked to the subsequent development of opioid dependency. Surgical patients are four-times more likely to be discharged with an opioid prescription compared to their nonsurgical counterparts [22,23]. Head and neck cancer patients are therefore at risk of developing chronic postoperative opioid use and previous research demonstrates up to 41% of patients continued to use opioids at 3-months postoperatively following primary surgical resection for oral cavity cancer [9].

Multiple surgical specialties have implemented postoperative MMA protocols in an effort to decrease the use of opioids in the postoperative period and the subsequent development of chronic postoperative use. A recent study in breast reconstruction patients by Astanehe et al. showed significantly reduced OMEs in an MMA managed cohort [16]. In Otolaryngology—Head & Neck surgery, MMA use in thyroid, parathyroid, and outpatient head and neck surgeries is safe and effective while concomitantly reducing opioid consumption [24,25]. There are few publications studying the use of MMA protocols in head and neck cancer surgery with free flap reconstruction. Vu et al. demonstrated reduced opioid use intra-operatively and in the post-anesthesia care unit (PACU) when patients were administered pre-operative oral celecoxib, gabapentin, and/or tramadol hydrochloride [13]. This study, however, did not investigate opioid use after the patients were discharged from PACU. Eggerstedt et al. implemented an MMA protocol for head and neck cancer patients undergoing free flap reconstruction and reported significantly less opioid use in the first 72-postoperative hours [26]. Our study expands on previous results in the literature by showing MMA is associated with less opioid consumption between PODs 1–14. The use of mixed-effects logistic regression in our analysis also adjusts for within-subject variation–something that has not been done in previous work. Our study, combined with previous evidence in the literature, supports the use of MMA as an opioid sparing strategy in managing postoperative pain in the head and neck cancer population.

MMA protocols used in other surgical populations minimize the use of opioids while providing stable and reliable pain control [27,28]. Arguably, head and neck cancer surgery with free flap reconstruction represents one of the most painful surgeries within Otolaryngology [29,30]. Our centre recently evaluated the effectiveness of postoperative pain management in a head and neck cancer surgery population and, despite the predominant use of opioids, 85% of our patients experienced ineffective pain control [4]. Eggerstedt et al. demonstrated the use of MMA in patients undergoing surgery for oral cavity cancer with free flap reconstruction is associated with improved pain control in both the immediate postoperative period and at discharge [26]. Similarly, in our study, pain control was better in the MMA group prior to POD 6. However, the pain control lines cross on POD 6 and the pre-MMA group appears to have better pain control from postoperative days 7–10. This result led us to explore why this might be occurring and two potential explanations were considered plausible. The first explanation is that patients in the pre-MMA group had increasing opioids prescribed as their hospital stay progressed and this could have resulted in lower pain scores overall. A second possible explanation pertains to the structure of the MMA protocol used at the time of this study. At the time of this study patients treated with the MMA protocol received scheduled acetaminophen, ibuprofen, and gabapentin for the first three postoperative days. After POD 3, all pain medications, including those provided as part of the MMA protocol, were prescribed on an as-needed basis. This change in schedule meant patients were not taking the MMA medications as frequently as when they were administered regularly; therefore, the analgesic effect, at least in some patients, would have become diminished. We, therefore, postulate that this is a potential reason why MMA patients did not have better pain control between PODs 7–10. Based on these results we have since adjusted our MMA protocol to extend the number of days we prescribe scheduled analgesics to POD 5. We believe by increasing the number of days of scheduled MMA dosing, we will have improved pain control compared to the results of this study.

The safety of MMA is a common concern among clinicians. Many clinicians believe NSAIDs are associated with an increased risk of postoperative bleeding. Several studies in the Otolaryngology literature show that with proper intraoperative hemostasis, perioperative NSAID use is not associated with increased postoperative bleeding [24,26,31]. Chin et al. did not demonstrate an increased risk of postoperative hematomas with the postoperative use of ketorolac for thyroidectomies [31]. In the study performed by Eggerstedt et al., they did not demonstrate any increased postoperative hematomas or flap failure in the MMA cohort [26]. Furthermore, a recent meta-analysis including 27 randomized clinical trials, did not show any increased risk of postoperative bleeding associated with NSAID use [32]. Acetaminophen is generally accepted to be safe in postoperative patients and effective in reducing opioid consumption; however, clinicians must use caution with the use of NSAIDs and acetaminophen in the head and neck cancer population because hepatic and renal dysfunction is more common within this population. We did not observe an increase in postoperative complications within our MMA cohort.

Our study is limited by its retrospective design that inherently includes bias secondary to patient selection. The MMA protocol was a component of a head and neck ERAS care pathway that also included preoperative education, management of intraoperative fluid balance, and intraoperative temperature control. The other ERAS care elements were also part of the ongoing CHERP protocol and we, therefore, do not believe the differences between ERAS and CHERP influenced the results.

This study is strengthened by its use of high-quality administrative data that was collected prospectively at the point of care. Such data are highly reliable, and we are confident in its accuracy and reliability. Another strength of this study is our analytical approach to the data which adjusted for variation within subjects and has improved our understanding of our patients’ postoperative pain experience using different treatment protocols. There are few published studies of MMA use in major head and neck cancer surgery with free flap reconstruction and none that incorporate the entire postoperative course. We have shown that MMA protocols are effective and are associated with reduced postoperative opioid consumption in head and neck cancer patients.

## 5. Conclusions

We conclude that in a tertiary academic head and neck surgical oncology program MMA is a feasible method of postoperative pain management in patients undergoing major head and neck cancer surgery with free flap reconstruction. The use of MMA is associated with a significant decrease in the use of opioids and potentially decreased side effects associated with opioid use.

## Figures and Tables

**Figure 1 cancers-13-01266-f001:**
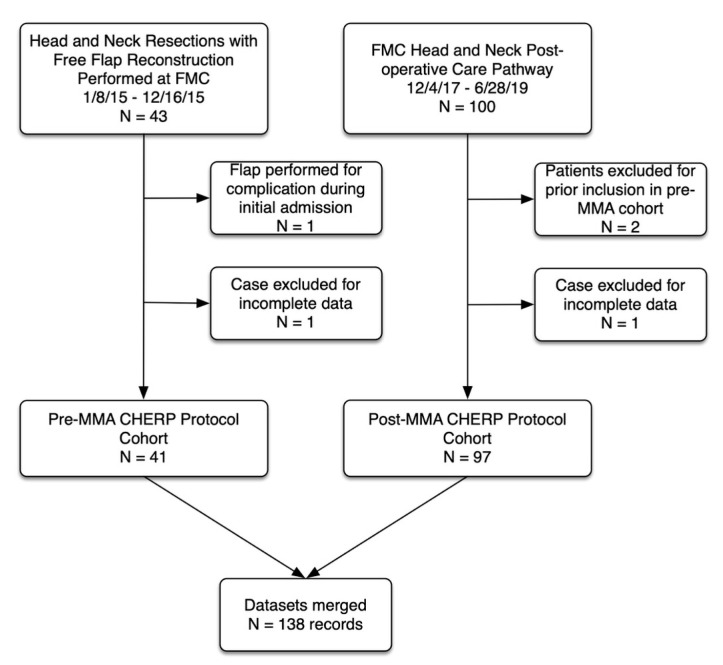
Study flow diagram. CHERP (Calgary Head and Neck Enhanced Recovery Program), MMA (multimodal analgesia).

**Figure 2 cancers-13-01266-f002:**
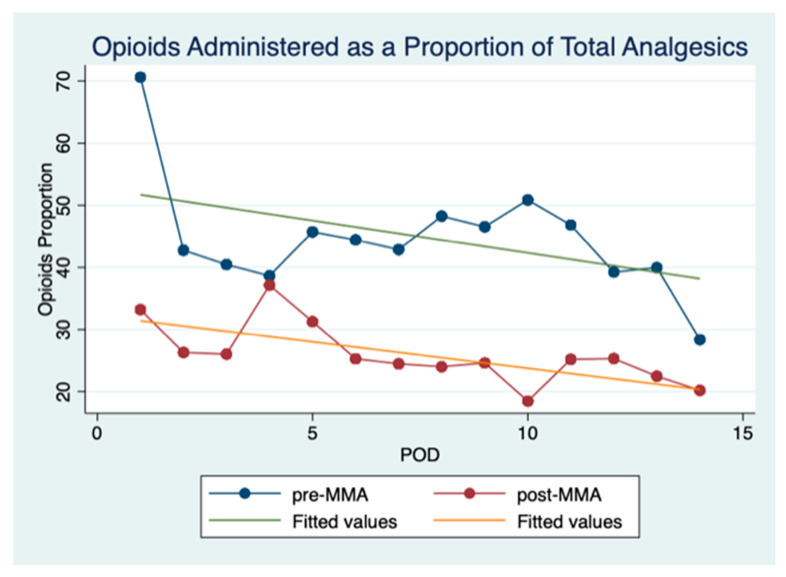
Opioids administered as a proportion of total analgesics. This figure shows opioids administered as a proportion of the total analgesics administered per POD (postoperative day).

**Figure 3 cancers-13-01266-f003:**
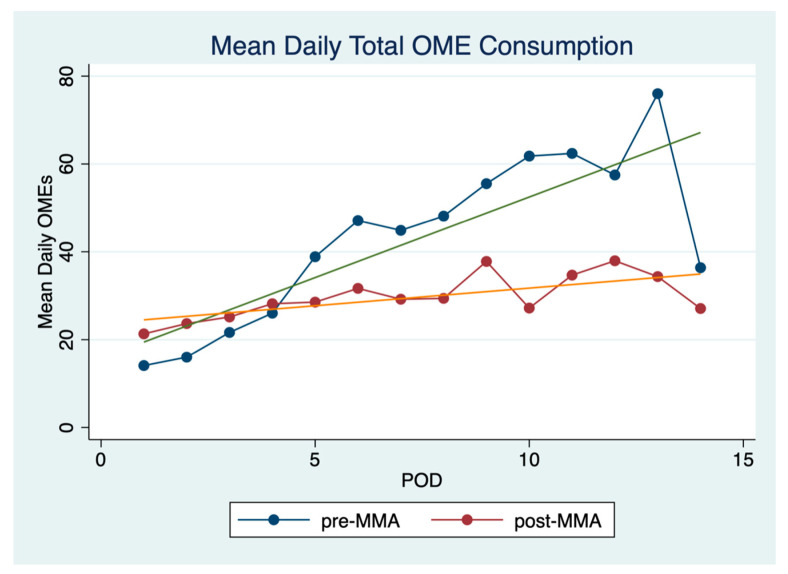
Mean daily OME consumption. Mean daily OME (oral morphine equivalents) consumption over the first 14 postoperative days.

**Figure 4 cancers-13-01266-f004:**
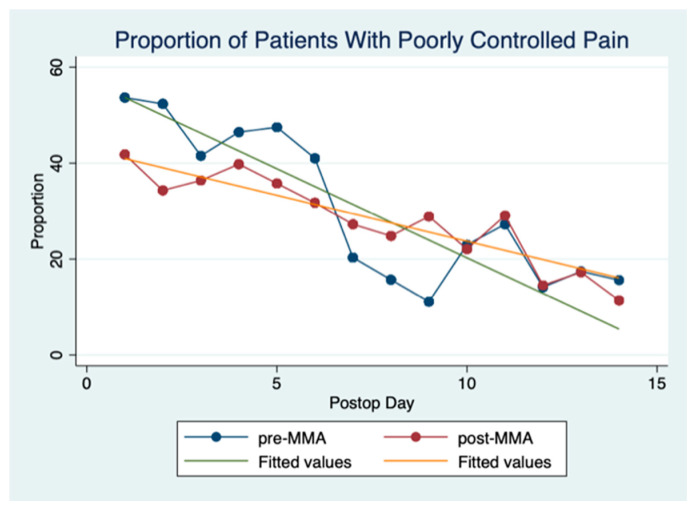
Proportion of pain scores >3 per POD. This figure shows the proportion of pain scores that are >3 per POD (postoperative day).

**Table 1 cancers-13-01266-t001:** Clinical Characteristics.

Characteristic	Number of Subjects (%)		*p*-Value
	Pre-MMA *n* = 41	Post-MMA *n* = 97	
Gender			ns
Male	32 (78%)	66 (68%)	
Female	9 (22%)	31 (32%)	
Age (year)			ns
Mean (SD)	61.2 (12.25)	61.9 (11.85)	
Range	23.7–82.0	33.4–87.0	
Alcohol Consumption			ns
Never Drinker	5 (12%)	16 (16%)	
Current Drinker	24 (59%)	60 (62%)	
Previous Drinker	2 (5%)	13 (13%)	
Not Reported	10 (24%)	8 (8%)	
Smoking Status			ns
Never Smoked	5 (12%)	24 (25%)	
Current Smoker	14 (34%)	33 (34%)	
Ex Smoker	18 (44%)	31 (32%)	
Not Reported	4 (10%)	9 (9%)	
Primary Site			ns
Oral Cavity	23 (56%)	55 (57%)	
Paranasal Sinus	3 (7%)	8 (8%)	
Skin	3 (7%)	13 (13%)	
Other Site	12 (29%)	21 (22%)	
pT Classification			ns
T1–T2	22 (54%)	36 (37%)	
T3–T4	14 (34%)	49 (51%)	
Other	5 (12%)	12 (12%)	
pN Classification			ns
N0	23 (56%)	45 (46%)	
N1	4 (10%)	9 (9%)	
N2 or greater	7 (17%)	28 (29%)	
Other	7 (17%)	15 ((16%)	
Clinical Stage			ns
Stage I-II	16 (39%)	20 (21%)	
Stage III-IV	20 (49%)	58 (60%)	
Other	5 (12%)	19 (19%)	
Flap Composition			
Soft Tissue	34 (83%)	74 (76%)	ns
Bone	6 (15%)	20 (21%)	
Soft Tissue + Bone	1 (2%)	3 (3%)	
Comorbidities			
Diabetes	5 (12%)	7 (7%)	ns
COPD	7 (17%)	7 (7%)	ns
Hypertension	18 (44%)	35 (36%)	ns
Heart Disease	8 (20%)	10 (10%)	ns

**Table 2 cancers-13-01266-t002:** Postoperative outcomes:

Characteristic	Number of Subjects (%)		
	Pre-MMA *n* = 41	Post-MMA *n* = 97	*p*-Value
Mobilization (day)			
Mean (SD)	2.5 (2.12)	1.7 (2.96)	<0.001
Range	1–12	0–28	
Length of Stay (day)			ns
Mean (SD)	11.6 (5.46)	12.7 (12.13)	
Range	4–29	4–99	
Complications (Clavien Classification)			ns
No Complication	15 (37%)	48 (49%)	
Grade I-IIIa	22 (54%)	28 (29%)	
Grade IIIb and IV	4 (10%)	11 (11%)	
Not Decumented	0 (0%)	10 (10%)	

## Data Availability

The data for this study are under the custodianship of Alberta Health Services (AHS) and are therefore unavailable for sharing. Data can be made available after an appropriate data sharing and access agreement is formally completed.

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
