# Peer review of "Efficacy of Multimodal Analgesia for Postoperative Pain Management in Head and Neck Cancer Patients"

_cancers, 2021, doi:10.3390/cancers13061266_

Round 1

Reviewer 1 Report

Good structured work. Due to the lack of literature so far could be of value to the readers despite the retrospective study design.

Details and differentiation about the type of free flap reconstructions are missing. It can be assumed that patients reconstructed with "only" a radial forearm free flap would be classified differently from patients reconstructed with a DCIA or osseomyocutaneous scapula free flap. Therefore some more information would be useful here.

Author Response

Surasit Prueksawanit/MDPI
Assistant Editor

Cancers

Dear Dr. Prueksawanit:

Re: Manuscript cancers-1100619

Thank you for forwarding the reviewer comments and suggestions. I have responded to each suggestion (see below) and modified the manuscript where appropriate. I have used “track changes” in MSWord to document the areas that were revised. I trust our responses meet with your approval and we hope to see our work published in the forthcoming special head and neck issue of Cancers.

Kind regards,

Joe

Joseph C. Dort MD

Ohlson Professor of Head & Neck Surgery

Cumming School of Medicine, University of Calgary

Reviewer 1:

Reviewer Comment:

Good structured work. Due to the lack of literature so far could be of value to the readers despite the retrospective study design.

Details and differentiation about the type of free flap reconstructions are missing. It can be assumed that patients reconstructed with "only" a radial forearm free flap would be classified differently from patients reconstructed with a DCIA or osseomyocutaneous scapula free flap. Therefore some more information would be useful here.

Author Response:

We thank the reviewer for his / her positive review of our work. We agree that understanding flap composition is important.  In table 1 instead of naming each specific flap, we classified the flaps according to whether they were soft-tissue, bone or soft-tissue + bone. This classification is much easier to interpret than naming each individual flap. We have therefore left the table “as is” and hope the reviewer agrees with our approach.

Reviewer 2 Report

This is a carefully conducted and good retrospective study looking at a very germane and important question of the utility of multimodal analgesia in head and neck surgery patients. This is an important topic and refreshing to see a paper on perioperative head and neck surgery since this area is not well covered in the literature. The paper is generally well written and of merit, however there are a few issues that need to be considered.

General points

The findings are to some degree overstated and should be put stated more accurately. For example the reduction of opioid side effects is suggested several times but there are no direct data to support it.

There are no opioid adverse effect data. This is an important omission since the main premise of the study from the introduction is a reduction in opioids is the main goal.

There is no mention of adverse effects of gabapentinoids nor of any of the potential issues of continuing use or misuse (BMJ. 2019 Jun 12;365:l2147. doi: 10.1136/bmj.l2147.Associations between gabapentinoids and suicidal behaviour, unintentional overdoses, injuries, road traffic incidents, and violent crime: population-based cohort study in Sweden. Molero, Larsson, D'Onofrio, Sharp, Fazel ).

Also, the efficacy of perioperative gabapentinoids has been called into question (Perioperative Use of Gabapentinoids for the Management of Postoperative Acute Pain: A Systematic Review and Meta-analysis. Verret M, Lauzier F, Zarychanski R, Perron C, Savard X, Pinard AM, Leblanc G, Cossi MJ, Neveu X, Turgeon AF; Canadian Perioperative Anesthesia Clinical Trials (PACT) Group. Anesthesiology. 2020 Aug;133(2):265-279.) and should probably warrant some discussion.

It would be interesting to see some data as to which analgesics each of the groups actually took on the post operative days.

Would a reduction of a mean of 10mg morphine per day be expected to have a major reduction in opioid-mediated side effects?

I am unclear to what extent the current study adds to the data expounded by Reference 26. This  paper (Eggerstedt et al 2019) also reports on opioids at discharge which is the key factors in the contribution of perioperative opioids to the ‘opioid crisis’.

Specific Points

Line 58

Apart from epidemic being inaccurate in this context perhaps with the current pandemic it would be better to refer to it as ‘opioid crisis’

Line 68

A 2017 publication is not ‘recently published’

Line 71

MMA may reduce opioid side effects

Line 102

Is there a chance of Bias given the pre-MMA data is from at least 2 years previously and overall management may have improved?

Line 113

Social habits?

Line 134-147

Unusual statistics

Figure 2 Surely the pre-MMA will always have a higher proportion since the gabapentinoids are less likely to be used and therefore merely reflect the difference in protocols.

Figure 3

Is there significance to the very similar opioid use at day 14?

Line 184

The use of ‘surprisingly’ intimates an expectation bias of the authors. Similarly the use of ‘expectedly’ in Line 245.

Line 188

Is what way different? Statistically the lines may be different but clinically, using this outcome, MMA is ‘better’ pre day 6 and pre-MMA ‘better’ after. Can either be described as being ‘better’ than the other?

Figure 4

Were the pain scores at rest? Are there any data comparing pain score on movement (breathing etc)?

Table 2

Is the difference in time to mobilisation clinically significant? Furthermore, I’m struggling to understand how the P value can be 0.001 when variability of both groups (but especially the MMA group) is so high. Also, no comment is made on the presumed outlier of 28 days before mobilisation in the MMA group.

Line 108-109

The have been no data given to support this statement.

Line 210-215

Surely to examine this problem on should look at the number of patients in the two groups that are being discharged on opioids and how many are still requiring opioids at a define time point later. The data here show that at day 14 both the consumption and the proportion of opioids are similar in both groups.

Line 226

And this study did not investigate use at discharge and subsequently.

Line 233-235

Difficult to say this without any side effect data.

Line 236

The term ‘Narcotics’ has not been used before this, probably more accurate to stay with the term ‘opioids’.

Line 255-257

I do not understand this second explanation. If these medications were available PRN and the pain scores increased, why did the patients not ask for/get given these medications? Are there data that shows the actual use of these medications by postoperative day? Also why did the PRN medications appear to ‘work’ for 3 days (day 3 to day 6) but then not so well subsequently?

Line 275

Only the Clavien-Dindo classifications are presented. What was the actual data for bleeding as a complication?

Author Response

Surasit Prueksawanit/MDPI
Assistant Editor

Cancers

Dear Dr. Prueksawanit:

Re: Manuscript cancers-1100619

Thank you for forwarding the reviewer comments and suggestions. I have responded to each suggestion (see below) and modified the manuscript where appropriate. I have used “track changes” in MSWord to document the areas that were revised. I trust our responses meet with your approval and we hope to see our work published in the forthcoming special head and neck issue of Cancers.

Kind regards,

Joe

Joseph C. Dort MD

Ohlson Professor of Head & Neck Surgery

Cumming School of Medicine, University of Calgary

Reviewer 2:

General Comment:

This is a carefully conducted and good retrospective study looking at a very germane and important question of the utility of multimodal analgesia in head and neck surgery patients. This is an important topic and refreshing to see a paper on perioperative head and neck surgery since this area is not well covered in the literature. The paper is generally well written and of merit, however there are a few issues that need to be considered.

Author Response:

We appreciate reviewer 2’s thorough review of our manuscript and for his / her generally positive comments. Reviewer 2’s specific concerns are listed, and addressed, below.

General Points:

Reviewer Comment 1:

The findings are to some degree overstated and should be put stated more accurately. For example the reduction of opioid side effects is suggested several times but there are no direct data to support it.

Author Response:

We agree with the reviewer that we present no data to directly support the issue of opioid-associated side effects and have modified the manuscript as follows:

  • Line 72 I’ve removed the “while simultaneously reducing…” part of the sentence.

However, in our concluding paragraph (lines 304, 305) we state that side effects may be potentially reduced and this, in our opinion, is not an overstatement and merely a potential outcome.

Reviewer Comment 2:

There are no opioid adverse effect data. This is an important omission since the main premise of the study from the introduction is a reduction in opioids is the main goal.

Author Response:

This comment follows from the previous one and we agree that we presented no adverse event data. However, our study was designed to evaluate opioid consumption and pain control. Adverse event data were not available and therefore not part of the study. We agree that such data are important and should be part of future research, ideally a prospective cohort study.

Reviewer Comment 3:

There is no mention of adverse effects of gabapentinoids nor of any of the potential issues of continuing use or misuse (BMJ. 2019 Jun 12;365:l2147. doi: 10.1136/bmj.l2147.Associations between gabapentinoids and suicidal behaviour, unintentional overdoses, injuries, road traffic incidents, and violent crime: population-based cohort study in Sweden. Molero, Larsson, D'Onofrio, Sharp, Fazel ).

Also, the efficacy of perioperative gabapentinoids has been called into question (Perioperative Use of Gabapentinoids for the Management of Postoperative Acute Pain: A Systematic Review and Meta-analysis. Verret M, Lauzier F, Zarychanski R, Perron C, Savard X, Pinard AM, Leblanc G, Cossi MJ, Neveu X, Turgeon AF; Canadian Perioperative Anesthesia Clinical Trials (PACT) Group. Anesthesiology. 2020 Aug;133(2):265-279.) and should probably warrant some discussion.

Author Response:

The reviewer raises the very important subject of gabapentinoids and, to be honest, this is an issue our team is struggling with. As noted earlier, we do not have adverse event data and so cannot report these outcomes.

However, gabapentinoids remain an important consideration. The first paper the reviewer mentions (Molero et al) is one I was familiar with and it was not considered relevant to our study. This is a population-based study and not a perioperative cohort. The adverse outcomes associated with gabapentinoids are serious but in this study were most likely to occur in the 15-24 year age group. This is an age group very unlikely to be impacted by head and neck cancer. Furthermore, the Molero study seemed to suggest that patients older than age 55 might actually have a lower incidence of the adverse events associated with gabapentinoids. The 55+ age group are more likely to be impacted by head and neck cancer.

The second paper (Verret et al) is more germane. This paper was not cited in our work because most of our manuscript was written by the time Verret et al was published and I just missed it. So, first of all I appreciate the reviewer bringing this to my attention. Our team has struggled with our ongoing inclusion of gabapentin for a number of reasons, and the Verret paper shows very conclusive evidence that gabapentinoids just don’t work. Our team is currently reviewing this paper and it will be discussed at our next team meeting. I believe we will decide to discontinue the use of gabapentin from our MMA protocol.

I would like to cite this paper but my resident (Dr. Hinther) uses a non-standard reference management software and I don’t believe we can add the reference to the Cancers-formatted version of our manuscript.

I don’t believe we can, or should, edit our manuscript because gabapentin (albeit at a low dose and short duration) was used. We believe that the low dose and short duration used make it very unlikely that harm was caused by gabapentin use. We agree with the reviewer though that it was likely ineffective.

Reviewer Comment 4:

It would be interesting to see some data as to which analgesics each of the groups actually took on the post operative days.

Author Response:

Thank you for this comment. We are not able to provide these data at this level of resolution.

Reviewer Comment 5:

Would a reduction of a mean of 10mg morphine per day be expected to have a major reduction in opioid-mediated side effects?

Author Response:

As noted earlier, we do not have adverse event / side effect data in our study and would prefer not to speculate on what a clinically meaningful difference might be. However, in my opinion an average reduction of 10 mg OME per day is potentially clinically meaningful and in some patients the reductions were much greater than this.

Reviewer Comment 6:

I am unclear to what extent the current study adds to the data expounded by Reference 26. This paper (Eggerstedt et al 2019) also reports on opioids at discharge which is the key factors in the contribution of perioperative opioids to the ‘opioid crisis’.

Author Response:

Eggerstedt et al 2019 was a very good paper and it partially addressed the issue of perioperative pain control after major head and neck surgery with free flap reconstruction. Our study expands our knowledge of this topic in that we studied the entire postoperative course up to day 14 after surgery. This allowed exploration of temporal trends over the course of hospitalization. We believe these aspects and new, and valuable, knowledge for readers and expands our understanding of perioperative opioid use.

Reviewer Comment 7:

Apart from epidemic being inaccurate in this context perhaps with the current pandemic it would be better to refer to it as ‘opioid crisis’

Author Response:

Thank you for your comment. We agree and have edited line 65 to substitute the term “crisis” as suggested by the reviewer.

Reviewer Comment 8:

A 2017 publication is not ‘recently published’

Author Response:

We agree with the reviewer and have edited line 73 appropriately.

Reviewer Comment 9:

MMA may reduce opioid side effects

Author Response:

Thank you for the suggestion. I have modified line 77 to say “may reduce opioid related side effects.”

Reviewer Comment 10:

Is there a chance of Bias given the pre-MMA data is from at least 2 years previously and overall management may have improved?

Author Response:

As the reviewer suggests, there is a chance of bias that is inherent in a retrospective cohort study such as this. However, we believe the chance of bias is minimal because there is no patient selection and because all patients were treated by the same team, using the same protocol, except for the addition of MMA.

Reviewer Comment 11:

Social habits?

Author Response:

Social habits refer to smoking and drinking habits. We have edited line 123 to be more clear about this.

Reviewer Comment 12:

Unusual statistics

Author Response:

We agree that the mixed effects logistic regression method is not commonly seen in our literature. However, Dr. Fick, co-senior author, senior biostatistician strongly believed this was the correct approach to analyze these complex data and that a more “traditional” analysis was flawed for this data set. Our approach and rationale are described in the Statistical Analysis section.

Reviewer Comment 13:

Figure 2 Surely the pre-MMA will always have a higher proportion since the gabapentinoids are less likely to be used and therefore merely reflect the difference in protocols.

Author Response:

Thank you for this comment. The daily proportions only tell part of the story and must be considered along with daily OME consumption. Gabapentin was only used for days 1-3 and would not be expected to affect proportion after that.

Reviewer Comment 14:

Is there significance to the very similar opioid use at day 14?

Author Response:

Thank you for this comment. The method used was designed to show temporal trends for the duration of hospitalization. The difference seen on day 14 was therefore not considered in isolation. The absolute difference in OMEs on day 14 (9 mg) was small but, in my opinion, clinically significant.

Reviewer Comment 15:

The use of ‘surprisingly’ intimates an expectation bias of the authors. Similarly the use of ‘expectedly’ in Line 245.

Author Response:

Thank you for pointing this out. I believe the word choices most likely reflect poor English usage on our part rather than any inherent bias. The word surprisingly was removed on line 195. The word “unexpectedly” was removed on line 256.

Reviewer Comment 16:

Is what way different? Statistically the lines may be different but clinically, using this outcome, MMA is ‘better’ pre day 6 and pre-MMA ‘better’ after. Can either be described as being ‘better’ than the other?

Author Response:

Thank you for your comment. Overall, I’m not sure that one protocol is “better” than the other. Prior to day 6 the MMA protocol appears to achieve superior pain control whereas from days 7-9 the pre-MMA pain scores are lower. The 2 protocols are equivalent from days 10 – 14. What we can say is that opioid consumption appears to be lower in the MMA group. The relative superiority of one protocol to the other is up to the reader to determine within the context of his / her own hospital and program setting.

Reviewer Comment 17:

Were the pain scores at rest? Are there any data comparing pain score on movement (breathing etc)?

Author Response:

The pain scores were taken at rest. There are no data comparing pain scores on movement. This is an excellent suggestion that could be incorporated into a future prospective study.

Reviewer Comment 18:

Table 2

Is the difference in time to mobilisation clinically significant? Furthermore, I’m struggling to understand how the P value can be 0.001 when variability of both groups (but especially the MMA group) is so high. Also, no comment is made on the presumed outlier of 28 days before mobilisation in the MMA group.

Author Response:

The mean difference between the 2 groups is almost 1 day. Based on what we know about surgical enhanced recovery, I believe this is clinically, as well as statistically, significant. I cannot explain the p value except to say that is what it is. In clinical data sets such as this there are outliers and in our analysis, we did not remove them. We chose to report all data and in this case the outlier makes the MMA mobilization data possibly appear worse than it actually is.

Reviewer Comment 19:

The have been no data given to support this statement.

Author Response:

The line numbers are not the same in my manuscript as the reviewers. I don’t know which statement is being referred to. I am sorry.

Reviewer Comment 20:

Surely to examine this problem on should look at the number of patients in the two groups that are being discharged on opioids and how many are still requiring opioids at a define time point later. The data here show that at day 14 both the consumption and the proportion of opioids are similar in both groups.

Author Response:

I agree with the reviewer that this would have been a good idea. However, we did not collect these data, nor can we acquire it. We believe our study, despite its limitations, makes a significant contribution to our understanding of an important problem in an under-studied patient population.

Reviewer Comment 21:

Line 226

And this study did not investigate use at discharge and subsequently.

Line 233-235

Difficult to say this without any side effect data.

Author Response:

Unfortunately, due to line number discrepancy, I cannot determine what specific statements the reviewer is referring to. I’m happy to consider changes where necessary and feasible. I certainly agree that we did not investigate use at discharge and subsequently. We have another manuscript (under review) that looks at long term opioid use after major head and neck surgery with flap reconstruction and are hoping to publish this in the near future. Data from that study (different patient group) show that long term opioid use is a major issue in this patient population. It would have been nice to refer to this study but it’s not yet published.

Reviewer Comment 22:

Line 236

The term ‘Narcotics’ has not been used before this, probably more accurate to stay with the term ‘opioids’.

Author Response:

I agree. This was an oversight. The term “narcotics” has been replace with “opioids” in lines 247 and 252.

Reviewer Comment 23:

Line 255-257

I do not understand this second explanation. If these medications were available PRN and the pain scores increased, why did the patients not ask for/get given these medications? Are there data that shows the actual use of these medications by postoperative day? Also why did the PRN medications appear to ‘work’ for 3 days (day 3 to day 6) but then not so well subsequently?

Author Response:

We appreciate this comment from the reviewer. We have proposed one possible explanation but we don’t really know why or how this happened. Either patients did not ask or were not offered medication. We believe there was a “wash out” period that might explain the lag between the protocol change and measured pain score difference. Since performing this study we have changed our “non-prn” MMA protocol to the first 5 days and will study its impact on pain scores. We are also undertaking education with our nursing staff on the head and neck unit with the goal of improving the quality of pain control for our patients.

Reviewer Comment 24:

Line 275

Only the Clavien-Dindo classifications are presented. What was the actual data for bleeding as a complication?

Author Response:

The Clavien-Dindo classification is a validated, and broadly accepted, method of classifying and reporting surgical complications. We did not report individual complications. However, there was no difference in bleeding between the 2 groups.

END OF COMMENTS FROM REVIEWERS

Round 2

Reviewer 1 Report

Over 3/4 of the presented cohort are patients reconstructed with myocutaneous free flaps ("soft tissue flaps").  This should be mentioned.

Therefor a comparison between the "soft tissue "and the "bone tissue" group may be of relevance in future studies. 

Reviewer 2 Report

The authors have provided a detailed comment to demonstrate their consideration of the points raised. Whilst I do not necessarily totally agree will all their comments, they have made reasonable and rational arguments to support their position.